# Does greater patient involvement in healthcare decision-making affect malpractice complaints? A large case vignette survey

Søren Birkeland[1,2], Marie Bismark[3]*, Michael J. Barry[4,5], Sören Möller[1,2]

**1** Open Patient data Explorative Network, Odense University Hospital, Odense, Denmark, **2** Department of Clinical Research, University of Southern Denmark, Odense, Denmark, **3** Melbourne School of Population and Global Health, The University of Melbourne, Melbourne, Australia, **4** Harvard Medical School, Boston, Massachusetts, United States of America, **5** Informed Medical Decisions Program, Massachusetts General Hospital, Boston, Massachusetts, United States of America

* mbismark@unimelb.edu.au

## Abstract

### Background

Although research findings consistently find poor communication about medical procedures to be a key predictor of patient complaints, compensation claims, and malpractice lawsuits ("complaints"), there is insufficient evidence to determine if greater patient involvement could actually affect the inclination to complain.

### Objectives

We conducted an experimental case vignette survey that explores whether greater patient involvement in decision-making is likely to influence the intention to complain given different decisions and consequences.

### Methods

Randomized, national case vignette survey with various levels of patient involvement, decisions, and outcomes in a representative Danish sample of men. We used prostate specific antigen (PSA) screening in men aged 45 to 70 years as the intervention illustrated in 30 different versions of a mock clinical encounter. Versions differed in the amount of patient involvement, the decision made (PSA test or no PSA test), and the clinical outcomes (no cancer detected, detection of treatable cancer, and detection of non-treatable cancer). We measured respondents' inclination to complain about care in response to the scenarios on a 5-point Likert scale (from 1: very unlikely to 5: very likely).

### Results

The response rate was 30% (6,756 of 22,288). Across all scenarios, the likelihood of complaint increased if the clinical outcome was poor (untreatable cancer). Compared with scenarios that involved shared decision-making (SDM), neutral information, or nudging in favor of screening, the urge to complain increased if the patient was excluded from decision-

**Data Availability Statement:** The data underlying the results presented in the study are available from the CERN repository (DOI: 10.5281/zenodo.4742237).

**Funding:** The project was funded by a grant of: EUR 40,000 from the Danish Health Insurance Foundation (award n. 17-B-0038) and EUR 5,700 from the Lilly & Herbert Hansen's foundation (award n. 100063). The funders had no role in study design, data collection and analysis, decision to publish, or preparation of the manuscript.

**Competing interests:** The authors have declared that no competing interests exist.

**Abbreviations:** PCa, Prostate cancer; PSA, Prostate-Specific Antigen (PSA; RedCap, Research Electronic Data Capture; SDM, Shared Decision Making.

making or if the doctor had nudged the patient to decline screening (mean Likert differences .12 to .16, p < .001). With neutral involvement or nudging in favor of intervention, the desire to complain depended highly on the decision reached and on the patient's course. This dependence was smaller with SDM.

## Conclusions

Greater patient involvement in decision-making appears to be associated with less intention to complain about health care, with SDM resulting in the greatest reduction in complaint likelihood.

## Introduction

Involving patients in health care decisions has deep roots in the ethical principle of self-determination, but also presupposes that patients actually want to receive information and participate in choosing interventions [1]. Many countries enforce patient involvement in health care decisions through a legal requirement to obtain informed consent prior to any procedure. Some clinicians and researchers have questioned this assumption. They contend that choice of interventions is fundamentally a clinician's responsibility and that, rather than participating in decision-making, patients look for provider authenticity and willingness to assume responsibility for their medical care [2,3]. From this perspective, malpractice liability tends to result from clinicians' failure to choose wisely on behalf of their patients. The onus is thereby placed entirely on clinicians to avoid malpractice liability, fueling the practice of 'defensive medicine' for fear of being criticized for failing to order tests or other interventions prior to a poor outcome [4]. This perspective is at odds with both patient-centered care and wise use of resources. In parallel, it is acknowledged that rates of compensation claims, patient complaints, and malpractice lawsuits (hereafter collectively referred to as "complaints") are higher within surgical specialties [5] and among patients who have experienced severe and preventable injuries while receiving health care [6] suggesting that higher risk medical procedures tend to correspond with higher rates of complaints.

While patients' wishes to take legal action may inherently vary with the outcomes of care, empirical studies recurrently report deficient communication about options, risks, and benefits to be an important predictor of complaints. Findings from an early study by Beckman et al. suggested that the decision to initiate malpractice litigation is often associated with poor delivery of information and lack of collaboration in the delivery of health care [7]. Similarly, Levinson and colleagues afterwards identified significant differences in communication behaviors of no-claims and claims physicians in primary care [8]. Braddock and colleagues in their secondary analysis of the same data concluded that patient involvement was not routinely practiced, leading to impaired patient-physician relationships, reduced patient adherence to medical regimens, and other quality-of-care concerns [9]. Later studies have supported the central role of poor communication in complaints about health care [10,11]. Correspondingly, it would be natural to hope that greater information sharing and patient participation in decision-making may prevent some complaints, but evidence is lacking [12].

The aim of our study was to explore how different approaches to patient involvement impact patients' intentions to initiate a complaint, while accounting for the decision made and the subsequent health outcomes of that decision. More specifically, we tested the null hypothesis that the mode of patient involvement in decision-making is not significantly associated with the initiation of a healthcare complaint.

## Materials and methods

### Study overview

We conducted a large national survey on a randomly selected sample of adult men in Denmark. To capture opinions from as wide a group of men as possible and ensure outcomes are generalizable to a larger proportion of the population, we chose to include participants irrespective of their illness experience. We randomized participants to case vignettes illustrating various levels of patient involvement in choices for or against the model intervention of PSA screening and with different decisions and health outcomes (please see below Fig 1 and 'Survey vignettes'). The survey development process has been described in further detail elsewhere and included feedback from a group of adult men in the public space followed by a separate review-and-feedback process involving a panel of patients with PCa experience [13]. The representativeness of our survey respondents was established in two previous studies, comparing socio-demographic, decision control preferences, and personality characteristics with Danish and international datasets [14,15].

### A study model on patient involvement in health care decision-making

Fig 1 illustrates different levels of provider-patient interactions leading to health care decisions.

The amount and framing of information varies across the different levels of patient engagement. Pertinent information from the clinician may be provided 'neutrally' without any further recommendations or advice on the right choice for the patient in question, or the decision may be influenced (or 'nudged') by the clinician's personal beliefs, preferences, or 'defensive' strategies intended to influence the patient's choice in a predictable way [2,16]. Shared decision-making (SDM) is a means of obtaining the patient's informed consent following a high-degree of involvement [17]. In SDM, patients communicate with clinicians about possible interventions, informed by the best available evidence about various options and attention to patients' personal preferences [18]. SDM requires provision of all relevant information on the risks and benefits of different options [18]. To achieve this, clinical situation-specific 'Decision Aids' are commonly used in conjunction with an in-person conversation with the health provider to accurately and systematically share unbiased information and explore the patient's preferences [18].

### Intervention—The clinical model

We used Prostate Specific Antigen (PSA) screening for prostate cancer (PCa) as our model 'intervention'. PCa is among the most common cancers and a leading cause of cancer death among men worldwide [19]. PSA screening is, however, controversial [20]. It detects many clinically insignificant tumors which would never have caused symptoms but also sometimes

| Spectrum of patient participation in decision-making | Information | No information | Information with nudging against intervention | Neutral information | Information with nudging in favor of intervention | Decision aid and discussion with health professional |
| | Decision | Health professional makes decision | Patient consent or no consent following information about procedure | | | Shared decision making |

**Fig 1. Levels of patient involvement in health care decision-making.**

misses the presence of a significant PCa [21]. The American Urological Association and US Preventive Services Task Force guidelines have suggested ages 55 to 69 to be the key age range for offering PSA to average-risk men while European urology guidelines, among others, recommend an individualised risk-adapted strategy for PSA screening in well-informed men aged over 50 years, and men aged over 45 years if they have a PCa family history [19,22]. Due to the high risk of adverse effects relative to the limited gain, PSA screening tends to be considered an individual ('preference sensitive') decision based on whether the possible benefits are deemed to outbalance the risks associated with the test and ensuing treatment [20]. PSA screening therefore provides a good model for investigating health care users' views on the decision making process. To align with the guidelines mentioned above, we focused on the age span 45 to 70 years [19,22].

## Survey vignettes

After introducing participants to the project, our survey presented each participant with a mock clinical encounter. We randomly assigned each participant to one of 30 scenarios with an identical core structure. The scenarios differed with respect to *a)* the style of decision-making about having or not having a PSA with five categories of information (cf. Fig 1), *b)* the decision made (if a PSA test was decided upon), and *c)* the outcome, with three scenarios based on a '*favorable*' outcome (no prostate cancer), an '*unfortunate*' outcome (advanced, non treatable prostate cancer), and an '*intermediate*' outcome (prostate cancer detected but succesfully treated). The scenarios reflected those encountered in clinical practice and drew on case scenarios reported in the literature [23,24]. For example, in one scenario, the patient chose not to have a test done after slight nudging *against* the PSA test ("[...] the doctor would suggest himself not to have a PSA test done [...]) and afterwards was diagnosed with a non-treatable prostate cancer. In other scenarios, the illustrated patient chose to have a test without the latter statement ('neutral information') or after slight nudging *in favor of* the PSA test ("[...] the doctor would suggest you have a PSA test done 'to be safe' [...]") and afterwards was diagnosed with a treatable prostate cancer. In yet another scenario, the patient engaged in SDM dialogue with the doctor, and afterwards chose not to have a PSA test and was later diagnosed with a non-treatable prostate cancer [13,25]. In Table 1, extracts from one version of the scenario are shown. The SDM scenarios all included a publicly available decision aid translated and adapted from Burford, Kirby, and Austoker [26].

**Table 1. Extracts from one case vignette version.**

"*Imagine that you are seeing your doctor for a 'health check'. The doctor asks a number of questions for symptoms such as shortness of breath, abdominal pain, etc. Your answer to all those questions is 'No'. The doctor also asks if there are any other issues to discuss. Your answer again is 'No'. Afterwards, the doctor does a stethoscope examination of your chest. He also does a blood pressure, heart rate check-up, and a manual abdominal examination and tells you that everything seems ok.*"

[...]

"*Your doctor tells you about a blood test for prostate cancer. It is called PSA. The doctor also informs you that it is a personal decision whether you want to have the test or not. Therefore, a guidance tool has been developed to help making the decision [...] The doctor hands out the tool and invites you to go through it. Afterwards, the doctor offers to talk to you to clarify questions etc.*"

[...]

"*After carefully going through the material, you have a conversation about the test with your doctor.*"
"*You decide NOT to have a PSA test done*"

[...]

"*It appears that you have prostate cancer. During the course of treatment, you and your family get increasingly worried and see your doctor several times.*"

[...]

"*Fortunately, it is possible to remove the cancer by surgery without any further complications.*"

Finally, we assessed participants' inclination to initiate a complaint about health care if subjected to the illustrated scenario using a Likert scale with the participant having to pick a whole number from 1, very unlikely, to 5, very likely. In Denmark, patients may complain about health care by initiating a malpractice suit (through a 'patient injury compensation organization') or by complaining to a state disciplinary board without claiming compensation, or may choose to do both. Participants therefore responded to two items: *"How likely is it that you would claim compensation*?' and *'How likely is it that you would complain about the doctor's care*?'. In our data analysis, a simple average of those two ratings was used to provide an overall estimate of the complaint likelihood. We decided not to investigate the two ratings separately, as the distinction between compensation claims and complaints may vary among countries. Participants also rated their ability to identify with the situation described in the scenario on a Likert scale from 1 (strongly agree) to 5 (strongly disagree).

## Recruitment of participants

We used a web-based survey (RedCap®) and invited a large random national sample of men living in Denmark through the Danish National authorities' web-portal for communication with citizens [14]. With due consideration to respecting people's right to not participate in our survey, we chose to send out only one reminder after 14 days and closed the survey after another 14 days. The first wave was launched January 24, 2019 and the second wave was launched March 7, 2019. Basic demographic characteristics of our sample have been reported before [14,15]. The average age of participants was 59.1 years (SD 7.3). Regarding marital status, 79.5% were living with a partner while 20.5% were not. Regarding participants' affiliation with the labor market, 65.8% were working, 34.0% were unemployed or retired, and 0.2% were students.

## Sample size and statistical analyses

When exploring the comparative impact of various approaches to patient involvement on participants' intentions to complain about health care, even a small difference between approaches could be important in clinics with many daily patient encounters. Nevertheless, we conducted a power analysis to estimate the sample required to detect a medium size change in complaint likelihood in response to the patient-physician encounter illustrated in the case vignettes. Our power calculations showed that 100 participants per group were needed to obtain 0.90 power to detect a 0.45 Likert score difference (that is, roughly half way from, e.g., 'complaint unlikely' to 'complaint somewhat likely') with a hypothesized standard deviation on the complaint likelihood rating of 1, an $\alpha$ level of .05, and a bi-directional, two-sample homoscedastic t-test [27,28]. We included an additional 300 participants in each of the 30 groups to compensate for non-responders (based on an expected response rate of 25% in on-line surveys of this kind with only one reminder [29,30]) and to compensate for expected non-normality of the measurements. We thereby planned to survey a total of 12,000 potential participants [31]. To address the risk of skewed response rates among groups we obtained permission to launch up to 3 waves of 12,000 invitations. Outcome measurements between groups were compared with linear regression with three different levels of exposure definition: the first five groups signified overall level of involvement, the next ten groups took into account the conduction of PSA or not, and the final 30 groups also took into account the clinical outcome. To compensate for non-normality of residuals as examined by quantile-quantile-plots, confidence intervals and p-values were determined by bootstrapping with 1000 repetitions. All statistical analyses were performed using Stata 16.

### Ethics approval and consent to participate

Under Danish law, research using questionnaires is exempted from ethical approval (Act on Research Ethics Review of Health Research Projects, Para 14). The launching of the survey, however, requires compliance with compliance with EU regulation 2016/679 and Directive 95/46/EC, General Data Protection Regulation (GDPR) as well as Danish Health Data Agency authorization (n. FSEID-00003692).

## Results

Thirty percent of invited men responded across two waves of invitations (Fig 2). The representativeness of the sample is described in prior papers [14,15]. Across the different levels of involvement, decisions and outcomes described in the scenarios, respondents felt able to identify with the scenario with only 6% responding negatively (Likert scores of 4 or 5) on this item. Overall, participants were 'unlikely' to initiate a complaint with a Likert rating mean of 2.08 (95% CI 2.06; 2.10). Irrespective of the level of patient involvement, respondents' intent to complain increased with less favourable course scenarios (as observed by an increase in Likert ratings reported, which reflect higher likelihood of complaining; please see Table 2, $P<0.001$ for all involvement/decision groups). Additionally, the inclination towards complaining generally was influenced by the decision made ($P = 0.004$). On the other hand, patient involvement in decision-making with the provision of neutral information, information in favor of intervention (PSA), and SDM resulted in a lower propensity to complain (second column in Table 2 with ratings aggregated across 'Decision' and 'Course' variants).

Actual differences with confidence intervals and standardized effect sizes using the 'no involvement', 'no PSA' and favorable (no prostate cancer) scenarios as the reference group are shown in Table 3. In scenarios with neutral information, nudging in favor of PSA, and SDM, the likelihood of a complaint was significantly reduced as compared to scenarios without any patient involvement in decision-making about PSA and scenarios illustrating nudging against PSA.

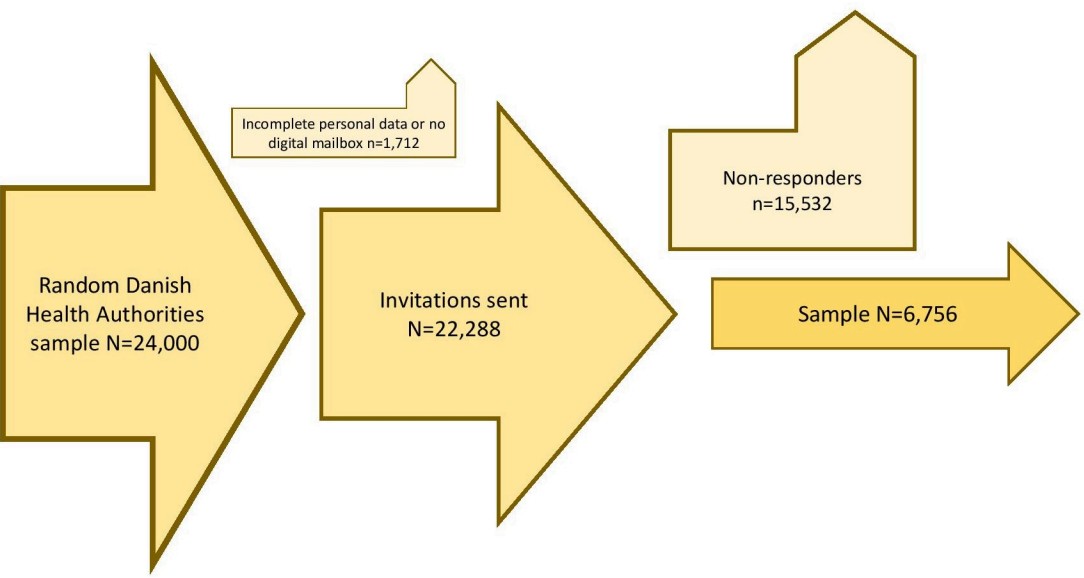

**Fig 2. Inclusion of study sample.**

**Table 2. Respondents' wish to complain, different levels of involvement, decisions and courses.**

| Involvement | Mean (95% CI) (*) N = 6,755 (***) | Decision | Mean (95% CI) (*) N = 6,755 | Course (**) | Mean (95% CI) (*) N = 6,755 | n |
|---|---|---|---|---|---|---|
| No involvement | 2.16 (2.11; 2.20) | No PSA | 2.12 (2.06; 2.19) | A (no PCa) | 1.76 (1.67; 1.85) | 238 |
| | | | | B (treatable PCa) | 2.13 (2.02; 2.24) | 222 |
| | | | | C (non treatable PCa) | 2.52 (2.41; 2.64) | 209 |
| | | +PSA | 2.19 (2.12; 2.25) | A (no PCa) | 1.84 (1.75; 1.93) | 267 |
| | | | | B (treatable PCa) | 2.26 (2.15; 2.37) | 244 |
| | | | | C (non treatable PCa) | 2.54 (2.42; 2.65) | 211 |
| Nudging against PSA | 2.20 (2.15; 2.25) | No PSA | 2.28 (2.21; 2.35) | A (no PCa) | 1.94 (1.83; 2.05) | 228 |
| | | | | B (treatable PCa) | 2.32 (2.20; 2.43) | 243 |
| | | | | C (non treatable PCa) | 2.58 (2.45; 2.70) | 230 |
| | | +PSA | 2.12 (2.05; 2.19) | A (no PCa) | 1.73 (1.64; 1.82) | 238 |
| | | | | B (treatable PCa) | 2.17 (2.06; 2.29) | 230 |
| | | | | C (non treatable PCa) | 2.48 (2.36; 2.60) | 217 |
| Neutral info (Reference) | 2.03 (1.99; 2.08) | No PSA | 2.03 (1.97; 2.10) | A (no PCa) | 1.74 (1.64; 1.84) | 225 |
| | | | | B (treatable PCa) | 2.01 (1.90; 2.12) | 233 |
| | | | | C (non treatable PCa) | 2.39 (2.26; 2.51) | 201 |
| | | +PSA | 2.03 (1.97; 2.10) | A (no PCa) | 1.72 (1.62; 1.82) | 219 |
| | | | | B (treatable PCa) | 2.06 (1.95; 2.16) | 230 |
| | | | | C (non treatable PCa) | 2.32 (2.20; 2.44) | 222 |
| Nudging for PSA | 2.00 (1.95; 2.04) | No PSA | 2.03 (1.97; 2.10) | A (no PCa) | 1.76 (1.66; 1.85) | 228 |
| | | | | B (treatable PCa) | 1.98 (1.87; 2.09) | 229 |
| | | | | C (non treatable PCa) | 2.35 (2.23; 2.48) | 230 |
| | | +PSA | 1.96 (1.90; 2.02) | A (no PCa) | 1.70 (1.61; 1.79) | 208 |
| | | | | B (treatable PCa) | 1.96 (1.86; 2.06) | 238 |
| | | | | C (non treatable PCa) | 2.21 (2.09; 2.33) | 220 |
| SDM | 2.00 (1.96; 2.05) | No PSA | 1.97 (1.91; 2.03) | A (no PCa) | 1.82 (1.73; 1.91) | 261 |
| | | | | B (treatable PCa) | 1.95 (1.84; 2.05) | 220 |
| | | | | C (non treatable PCa) | 2.19 (2.08; 2.30) | 210 |
| | | +PSA | 2.03 (1.96; 2.10) | A (no PCa) | 1.88 (1.78; 1.99) | 196 |
| | | | | B (treatable PCa) | 1.99 (1.87; 2.10) | 214 |
| | | | | C (non treatable PCa) | 2.24 (2.11; 2.37) | 194 |

* '1' means complaint very unlikely and 5 a complaint is very likely.

** PCa: Prostate cancer.

*** N amounts to 6,755 as one participant did not respond to complaint likelihood items.

The variation across decision and course variants is illustrated in Fig 3. In scenarios that involved a poor outcome (untreatable cancer), participants who had engaged in SDM were the least likely to express an intention to complain.

## Discussion

Patient involvement and considerations about an individual's right to autonomy is a common discussion in bioethics [1,32]. Moreover, patient-centered care has become an essential part of delivering quality care in modern medicine. For example, the US Committee on the Quality of Health Care recommended that health care in the 21st century should be patient-centred and respectful of individual patient preferences, needs, and values [33]. Similarly, involving patients in health care decisions is increasingly required by law, medical codes, and standards with SDM providing a promising approach towards making real the intentions of patient-

**Table 3. Group differences in respondents' wish to complain, different levels of involvement, decisions and courses.**

| Involvement | Difference (95% CI)(*) N = 6,755 (***) | Decision | Difference (95% CI) (*) N = 6,755 | Course (**) | Difference (95% CI) (*) N = 6,755 |
|---|---|---|---|---|---|
| No involvement | (Reference) | No PSA | (Reference) | A (no PCa) | (Reference) |
| | | | | B (treatable PCa) | 0.37 (0.23; 0.51) P<0.001 |
| | | | | C (non treatable PCa) | 0.76 (0.62; 0.91) P<0.001 |
| | | +PSA | 0.07 (-0.03; 0.16), P = 0.159 | A (no PCa) | 0.08 (-0.04; 0.21) P = 0.198 |
| | | | | B (treatable PCa) | 0.50 (0.36; 0.65) P<0.001 |
| | | | | C (non treatable PCa) | 0.78 (0.63; 0.92) P<0.001 |
| Nudging against PSA | 0.04 (-0.02; 0.11) P = 0.203 | No PSA | 0.16 (0.06; 0.25), P<0.001 | A (no PCa) | 0.18 (0.03; 0.32) P = 0.016 |
| | | | | B (treatable PCa) | 0.56 (0.42; 0.70) P<0.001 |
| | | | | C (non treatable PCa) | 0.82 (0.67; 0.97) P<0.001 |
| | | +PSA | -0.00 (-0.10; 0.09), P = 0.927 | A (no PCa) | -0.03 (-0.16; 0.10) P = 0.646 |
| | | | | B (treatable PCa) | 0.41 (0.27; 0.56) P<0.001 |
| | | | | C (non treatable PCa) | 0.72 (0.57; 0.87) P<0.001 |
| Neutral info | -0.12 (-0.19; -0.06) P<0.001 | No PSA | -0.09 (-0.18; 0.00), P = 0.062 | A (no PCa) | -0.02 (-0.15; 0.11) P = 0.763 |
| | | | | B (treatable PCa) | 0.25 (0.12; 0.38) P<0.001 |
| | | | | C (non treatable PCa) | 0.63 (0.47; 0.78) P<0.001 |
| | | +PSA | -0.09 (-0.18; 0.00), P = 0.062 | A (no PCa) | -0.04 (-0.17; 0.09) P = 0.543 |
| | | | | B (treatable PCa) | 0.30 (0.16; 0.43) P<0.001 |
| | | | | C (non treatable PCa) | 0.56 (0.41; 0.71) P<0.001 |
| Nudging for PSA | -0.16 (-0.22; -0.09) P<0.001 | No PSA | -0.09 (-0.18; 0.00), P = 0.063 | A (no PCa) | -0.00 (-0.13; 0.12) P = 0.952 |
| | | | | B (treatable PCa) | 0.22 (0.08; 0.37) P = 0.002 |
| | | | | C (non treatable PCa) | 0.59 (0.44; 0.75) P<0.001 |
| | | +PSA | -0.16 (-0.25; -0.07), P<0.001 | A (no PCa) | -0.06 (-0.19; 0.07) P = 0.362 |
| | | | | B (treatable PCa) | 0.20 (0.06; 0.33) P = 0.004 |
| | | | | C (non treatable PCa) | 0.45 (0.30; 0.60) P<0.001 |
| SDM | -0.16 (-0.22; -0.09) P<0.001 | No PSA | -0.15 (-0.24; -0.06), P = 0.001 | A (no PCa) | 0.06 (-0.07; 0.19) P = 0.378 |
| | | | | B (treatable PCa) | 0.19 (0.05; 0.32) P = 0.007 |
| | | | | C (non treatable PCa) | 0.43 (0.28; 0.58) P<0.001 |
| | | +PSA | -0.09 (-0.18; 0.00), P = 0.063 | A (no PCa) | 0.12 (-0.01; 0.26) P = 0.074 |
| | | | | B (treatable PCa) | 0.23 (0.08; 0.37) P = 0.002 |
| | | | | C (non treatable PCa) | 0.48 (0.32; 0.63) P<0.001 |

* '' means complaint very unlikely and 5 a complaint is very likely.

** PCa: Prostate cancer.

*** N amounts to 6,755 as one participant did not respond to complaint likelihood items.

centeredness. At the same time, a growing body of evidence suggests that SDM supported by patient decision aids improves the quality of health care decisions [34]. If sharing of health care decisions can also reduce patients' propensity to complain in the wake of a poor outcome, this finding might add further weight to efforts to engage patients in decisions. However, the

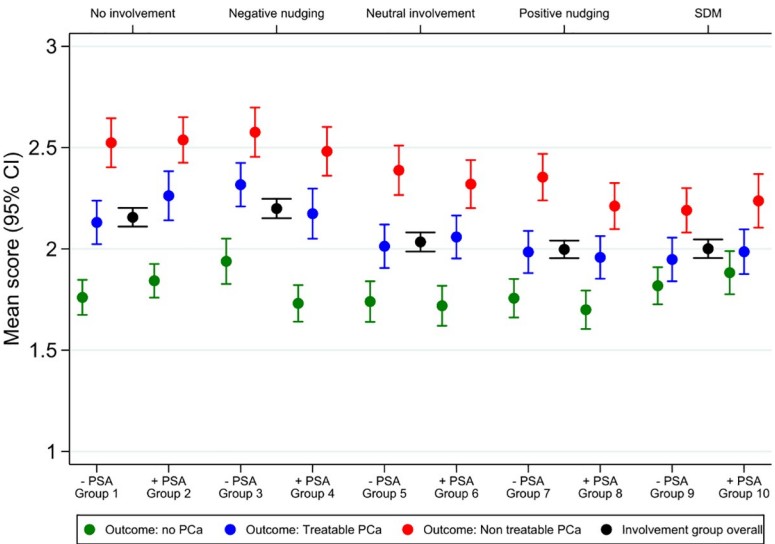

**Fig 3. The likelihood that survey respondents would complain.**

evidence is scant [7,11,12,35]. In a 2015 systematic review of the literature, Durand and colleagues concluded that "*there is insufficient evidence to determine whether or not shared decision-making and the use of decision support interventions can reduce medical malpractice litigation*" and that further investigation was required [12]. Since this systematic review was published, there has only been one further study in this area to our knowledge. A crowd-sourcing simulation study among adult US citizens by Schoenfeld et al. suggested that when comparing SDM with no SDM, the odds ratio for contacting a lawyer was 0.20 (95% CI 0.12 to 0.31) [36]. In that study of 804 participants, vignettes representative of SDM illustrated concise verbal delivery of neutral information from a doctor without the use of decision aids. Furthermore, decisions were fixed (no CT scan for acute abdominal pain) as was the patient outcome (ruptured appendicitis). The present study aims to take into account these important factors to provide a long-awaited piece of evidence regarding the influence of various levels of patient involvement, decisions made, and clinical outcomes on health care users' urge to complain.

## Interpretation of findings

Overall, respondents were unlikely to express an intent to initiate a complaint against their clinician (Likert ratings 2.00 to 2.20). However, scenarios which lacked patient involvement were generally associated with a greater desire to complain than scenarios where the patient received plain, neutral information and participated in decision-making (Likert rating difference 0.12; 95% CI -0.19; -0.06; P<0.001). It should be kept in mind that while some of the scenarios included a poor outcome, or decisions made without properly informed consent, there was no indication of negligent care. The question could be posed why some respondents would express an intent to complain anyway, particularly when presented with a "positive outcome" scenario? In the real world, sizeable numbers of complaints are filed for care that is ultimately found to be non-negligent [11,37,38]. In this regard, our analyses, keeping everything but the communication about health care decisions invariable, suggests that, although far from being the only factor causing complaints, poor communication may constitute a key driver for patients to complain about health care. Findings agree well with previous research showing that patients are less likely to sue their clinician for medical malpractice when the provider

displays person-centered communication skills [8,39]. Our findings also suggest that, from the perspective of malpractice complaint risk, SDM is superior to simply providing "neutral information" (Likert rating difference -0.16 compared to -0.12). There may be many explanations for this result. For example, SDM may create awareness about risks that patients prioritize higher than clinicians and those risks may not be sufficiently addressed through neutral information. In other words, the risks patients are interested in are not always the ones that practitioners routinely disclose [40].

Furthermore, respondents tended to be positive towards PSA screening (groups 4, 6 and 8; Likert rating differences -0.00, -0.09, and -0.16) and nudging in favor of screening (group 8). When using nudging, clinicians may downplay the risk of interventions or explicitly favor ordering tests and treatments without exploring the views of their patients [12,41]. This may occur for one of two reasons. First, they may believe that their own clinical recommendation will achieve better outcomes than involving patients in the decision-making. Secondly, as noted by Studdert and colleagues, a culture of "defensive medicine" encourages the ordering of excessive tests and procedures in the belief that these practices reduce the risk of lawsuits [4]. Viewing from a medico-legal perspective, in our survey, scenarios with PSA screening without patient involvement (group 2) and nudging towards PSA screening 'to be safe' (group 7 and 8) could be considered representative of such a defensive approach. Findings suggest that defensively applying the intervention (PSA) without any patient involvement is associated with an increased risk of complaint if the outcome is poor (Likert rating difference 0.78; 95% CI 0.63–0.92; P<0.001). Use of nudging towards an intervention, however, resulted in complaint likelihood ratings similar to the provision of neutral information or use of SDM. Interestingly, the inclination to complain was most predictable (homogenous ratings) across outcomes with SDM. In other words, the physician's risk of facing a malpractice complaint seems less dependent on decisions reached and the patient's later course when applying SDM. A strategy of "positive nudging" was associated with slightly (but not significantly) lower intent to complain in cases where the test was done and no cancer occurred (Likert rating difference -0.06; 95% CI -0.19–0.07; P = 0.362). This suggests that defensive medicine's frail protection from medico-legal events arises not from success in preventing harm (such as by detecting treatable cancers), but from the large number of interventions, with their inherent drawbacks, that occur in patients who would have remained well regardless.

## Limitations

Our study has several limitations. First, the study only investigates the impact of patient involvement on malpractice complaints in men. Similarly, we must acknowledge the limitation and possible lack of generalizability of a study done in a single country. However, Danish men's reactions to the scenarios we present may generalize to men in other Western countries given the international controversy about the appropriate role of PSA screening, even if the malpractice climate differs from country to country [22]. The scenarios presented to participants certainly involve physician behaviors that might be seen in the United States, for example. Secondly, the possibility of non-response bias must be mentioned. As is the case with other surveys, responders to our survey may not accurately represent the entire population of men aged 45 to 70 years. We recruited participants through a national web-based communication channel [14]. Findings from previous studies on our sample's representativeness do not indicate that an important selection bias was introduced through this approach [14,15]. Nevertheless, even if our sample is representative, we cannot fully rule out the possibility of a residual response bias [14,15]. Similarly, analyses do not allow for conclusions about the subgroup of participants with personal experience with PCa. Some further limitations regarding

respondents' assessment of the hypothetical scenarios deserve mention. Case vignettes have been successfully used in previous studies of health care decision-making in prostate cancer [23,42] but it is important to keep in mind that participant responses to vignettes reflect hypothetical judgements. Still, pilot-testing during survey development as well as ratings from the survey itself suggest that respondents felt able to identify with the patients in the situations described in vignettes [13].

Despite these limitations, our findings seem to correspond well with prior research on the role of communication in malpractice complaints [7,8,11,12,39]. Strengths of our study include the large sample size, and the ability to control for important variables including the decision made about having the procedure in question and the ultimate outcome, which have not been controlled for in previous research [36].

## Conclusions

Achieving the vision of patient-centred care is widely viewed as a cornerstone to high quality healthcare. Engaging patients in decision-making has many potential benefits: it respects individual autonomy, allows patients to weigh risks that matter to them personally, and may help to align patient expectations with realistic outcomes taking into account situation-specific risks and individual concerns. In this regard, SDM can be seen as an evolution and extension of informed consent, involving not only providing accurate information about an intervention's harms, benefits, and alternatives, but also meaningful patient involvement in making a decision to proceed. In turn, SDM may help to avert the 'preventable' medico-legal aftermaths that arise from poor communication and failure to engage patients in decisions that have far-reaching consequences. Findings from our study suggest that greater patient involvement in health care decision-making may indeed provide some protection against complaints, even when the outcome is poor. 'Defensive' approaches, such as nudging patients towards an intervention 'to be safe', may also provide some protection although the apparent protection predominantly arises from a reduced complaint likelihood as observed by Likert scores in courses of health care where patients underwent an intervention and would have remained well in any case. Overall, greater patient involvement such as SDM achieved the most consistent reduction in intent to complain regardless of the decision made or the eventual health outcome.

## Supporting information

**S1 File. English translation of Danish survey.** Informed choice survey.
(PDF)

**S2 File. PSA test for prostate cancer.** Decision aid.
(PDF)

**S3 File. Undersøgelse om patienters selvbestemmelse.** Danish version survey.
(PDF)

**S4 File. PSA-test for prostatakræft.** Danish version decision aid.
(PDF)

**S5 File. STROBE statement-checklist: Does greater patient involvement in healthcare decision-making affect malpractice complaints?–A large case vignette survey.** STROBE.
(PDF)

## Acknowledgments

We would like to thank survey respondents for their participation and invaluable contributions to the current study.

## Author Contributions

**Conceptualization:** Søren Birkeland, Michael J. Barry.

**Data curation:** Søren Birkeland.

**Formal analysis:** Søren Birkeland, Sören Möller.

**Funding acquisition:** Søren Birkeland.

**Investigation:** Søren Birkeland.

**Methodology:** Søren Birkeland, Michael J. Barry, Sören Möller.

**Project administration:** Søren Birkeland.

**Validation:** Søren Birkeland, Marie Bismark, Michael J. Barry, Sören Möller.

**Writing – original draft:** Søren Birkeland.

**Writing – review & editing:** Marie Bismark, Michael J. Barry, Sören Möller.

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
