## [Decision Letter · Decision Letter 0]

17 Mar 2021

PONE-D-20-40761

Does shared decision-making in healthcare reduce malpractice complaints?

PLOS ONE

Dear Dr. Birkeland,

Thank you for submitting your manuscript to PLOS ONE. After careful consideration, we feel that it has merit but does not fully meet PLOS ONE’s publication criteria as it currently stands. Therefore, we invite you to submit a revised version of the manuscript that addresses the points raised during the review process.

We look forward to receiving your revised manuscript.

Kind regards,

Ritesh G. Menezes, M.B.B.S., M.D., Diplomate N.B.

Academic Editor

PLOS ONE

Journal Requirements:

2. In your Methods section, please provide additional information about the participant recruitment method and the demographic details of your participants. Please ensure you have provided sufficient details to replicate the analyses such as: a) the recruitment date range (month and year), b) a table of relevant demographic details.

Reviewers' comments:

Reviewer's Responses to Questions

**Comments to the Author**

1. Is the manuscript technically sound, and do the data support the conclusions?

Reviewer #1: Yes

Reviewer #2: Yes

Reviewer #3: Yes

Reviewer #4: Yes

Reviewer #5: Yes

Reviewer #6: Yes

2. Has the statistical analysis been performed appropriately and rigorously? 

Reviewer #1: No

Reviewer #2: I Don't Know

Reviewer #3: Yes

Reviewer #4: Yes

Reviewer #5: I Don't Know

Reviewer #6: I Don't Know

3. Have the authors made all data underlying the findings in their manuscript fully available?

Reviewer #1: No

Reviewer #2: No

Reviewer #3: No

Reviewer #4: Yes

Reviewer #5: Yes

Reviewer #6: No

4. Is the manuscript presented in an intelligible fashion and written in standard English?

Reviewer #1: Yes

Reviewer #2: Yes

Reviewer #3: Yes

Reviewer #4: Yes

Reviewer #5: Yes

Reviewer #6: Yes

5. Review Comments to the Author

Reviewer #1: Thanks to the authors for the opportunity to read this manuscript. I give my comments:

1. Abstract. It should be clearly indicated that these were mock clinical encounter using 30 scenarios.

2. Abstract. The authors write „…with various levels of patient involvement, decisions, and outcomes…”. Please add information about the decisions the authors have in mind.

3. Abstract. „With neutral involvement or nudging in favor of intervention, the desire to complain depended highly on the decision reached and on the patient’s course” – please explain what decision the authors mean.

4. Abstract. „…even in worst case scenarios” – please provide the result that supports this conclusion.

5. Introduction. The beginning of the introduction is very interesting, so it's a pity that the introduction is not longer. I am convinced that readers other than me would be equally eager to explore this issue. Please consider indicating at the end of the introduction the tested hypotheses.

6. Please consider including the first part of the discussion in the introduction.

7. Introduction. „The aim of our study was to explore the likely impact of patient involvement on health care users’ intentions to initiate a malpractice complaint given different approaches to patient involvement, while taking into account decisions made and healthcare outcomes.” – again, please explain what decision the authors mean.

8. Introduction. What other factors that influence a patient's complaints tendency have been investigated so far? How does the presented research complement this state of knowledge? In my opinion, the justification for this subject should be better justified.

9. Methods. No indication of the data collection period.

10. Please describe the mock clinical encounter in more detail

11. Methods. How many patients were assigned to specific scenarios? Did these groups differ in terms of sociodemographic variables?

12. Please consider including the sections „THEORETICAL FRAMEWORK: PATIENT INVOLVEMENT IN HEALTH CARE DECISIONMAKING” and „INTERVENTION – THE CLINICAL MODEL” in the introduction.

13. In my opinion the content of the "STATISTICAL ANALYSIS" section should mostly be found in the methods as a description of the study design / study size.

14. In the "STATISTICAL ANALYSIS" section, there is no indication of what statistical program was used, what tests were performed, what significance level was used.

15. Results. „Irrespective of the level of patient involvement, respondents’ intent to complain increased with less favourable course scenarios (Likert ratings going up; please see Table 1)” - was it statistically significant?

16. Results. „Additionally, accross all types of decision-making, the inclination towards complaining was influenced by the decision made” - please provide the result of the statistical analysis supporting this claim.

17. As required by the journal, access to more detailed data should be provided, e.g. in addition to summary statistics, there should be data points behind medians.

Reviewer #2: I thank the authors for the very interesting read. The manuscript identifies a gap in the literature on shared decision-making (SDM) in clinical settings in that it points out that there is insufficient data on the correlation between SDM and the prevalence of malpractice complaints. The authors addressed this gap by conducting a web-based survey with a random national sample of men living in Denmark. Survey participants were presented with one of 30 vignettes describing hypothetical scenarios of experiences with prostate specific antigen screening (PSA). The vignettes differed in the level of patient-involvement, whether a PSA was undertaken, and clinical outcome (whether the participant developed prostate cancer and whether they were successfully treated). The survey elicited participants’ stated desire to seek compensation or complain about the physician’s care for the scenario with which they were presented. The manuscript’s main contribution lies in analyzing the relationship between different hypothetical levels of patient involvement in PSA screening and participants’ stated desire to complain.

Overall, the manuscript’s structure is well constructed and the text is clearly written. The current state of the literature is reviewed concisely, yet sufficiently. The chosen methodology is sound and appropriate for purpose and the presented approach builds on previous research. The results presented in the manuscript are relevant and add to the discipline’s body of knowledge. The study’s limitations are addressed clearly. Some minor issues remain which can easily be addressed by the authors.

1) Abstract: “Conclusions: Involving patients in decision-making appears to offer some prevention against malpractice complaints with patients engaged in SDM less likely to complain even in worst case scenarios”

The phrasing is unfortunate as it presents malpractice complaints simply as something to be prevented, regardless of their merit. The phrase could be misread as saying that the likelihood of a patient complaining bears no relation to the quality of care received. The argument presented in the main text is much more nuanced and avoids these misinterpretations. I suggest rephrasing.

2) P.3: “Some clinicians and researchers have questioned this assumption. They contend that choice of interventions is fundamentally a health care provider’s responsibility and that, rather than participating in decision-making, patients look for provider authenticity and willingness to assume responsibility for their medical care (2, 3). From this perspective, the onus is on clinicians to avoid malpractice liability, fueling the practice of ‘defensive medicine’ for fear of being criticized for failing to order tests or other interventions prior to a poor outcome (4)”

It is not immediately clear how the second sentence follows from the first. The incentive structure for clinicians might be shaped as described in the latter sentence, but the authors need to spell out how this results from the mentioned shift in perspective.

3) p.3: “While patients’ wishes to take legal action may inherently vary with the outcomes of care, empirical studies recurrently report deficient communication about options, risks, and benefits to be an important predictor of malpractice complaints (5-7). Correspondingly, it would be natural to hope that greater information sharing and patient participation in decision-making may prevent some complaints, but evidence is lacking (7)”

This is an important argument that connects the quality of received care to the likelihood to complain. Some version of it should be added to the abstract to compliment the clinician-centered perspective presented there.

4) p.3: “STUDY OVERSIGHT” should read “study overview”

5) p.3.: “We conducted a large national survey on a randomly selected sample of adult men in Denmark”

Decision to elicit preferences of the general population instead of the affected should briefly be motivated.

6) p.3: “The survey was developed with patient and public involvement from the gender and age group of interest”

Please briefly expand on the methodology used for patient involvement.

7) P.5: “We used a web-based survey (RedCap®) and invited a large random national sample of men living in Denmark through the Danish National authorities’ web-portal for communication with citizens (9)”

Add a brief discussion of potential selection effects here.

8) P.8: Figure 3: Light yellow on white extremely hard to read, consider changing color scheme.

9) P.9: The first paragraph of the discussion section should be integrated into the background section.

10) P.9-11 Just a musing but it might be worthwhile to conceptualize patient involvement as an important aspect of good quality care rather than as something that happens independent from the health intervention proper. This may or may not be worth mentioning in the conclusion. In any case, it would provide a good counter-balance to the physician-centered view currently starring prominently in the manuscript.

Reviewer #3: The submitted manuscript describes a large national survey on a randomly selected sample of adult men in Denmark. The studied aimed to analyze whether higher levels of patient involvement in decision-making about healthcare interventions are likely to prevent malpractice complaints. The manuscript is well written and the methodology of the study is well described. The findings are relevant and well described in relation to previous studies. No fundamental objections to this manuscript, but I would recommend to expand the introduction, which is too short at this moment.

It is unclear from the manuscript whether the data is publicly available. Considering that it is an anonymous dataset, it shouldn’t be a problem to deposit the dataset.

Reviewer #4: The paper deals with a very relevant and up-to-date question of the effects of shared decision-making (SDM) on the patients’ inclination to complain or sue healthcare providers. While it is often presumed that there is a negative correlation between the two phenomena, it is yet to be proven. The paper provides a valuable contribution to the recent discussion which is not only very interesting for other researchers but also potentially practical for healthcare providers, health professionals, and various policymakers. For the reasons listed here and below, the reviewer recommends acceptation of the paper (perhaps with very minor revisions).

The methods used are appropriate for the aim of the study. Due to its statistically limited benefits, relatively high risk of adverse effects, and the possible grave negative consequences of false-negative results, the Prostate Specific Antigen screening seems to be a very suitable model of medical intervention for the purposes of the study. As far as the reviewer is qualified to assess statistical analysis’s adequacy, the collection, analysis, and interpretation of the data were accurate. The figures and tables are clear, readable, and support the results of the study.

The respondent sample is big enough to draw a conclusion. Technical aspects regarding the representativeness of the sample, as well as the patient and public participation in the development of the survey and some other data, are described and evaluated in already-published papers in different journals, which are referred to in the reviewed paper.

Inclusion of three variables (i.e. not only patient participation in decision-making but also the decision made by the patient and the outcome) and analysis of their interrelations arguably made the results more reliable since in the real world, the said variables (among others) undoubtedly affect the patient’s decision to complain. Nevertheless, the study is still limited by its exclusive use of hypothetical judgments, which are arguably not influenced by emotions related to one’s own (perceived) health harm. Even though the respondents reported feeling able to identify with the patients in the case scenarios, further research would be necessary to establish the importance of emotions and other aspects of decision-making regarding complaining or suing healthcare providers in the real world. Nevertheless, this and several other limitations are addressed in the paper in a way that can be considered sufficient.

There are just two minor issues:

- Rarely there are typing errors (e.g. “litterature on p. 5 or “coloumn” on p. 6). It is only a marginal problem, but another proofreading might be useful.

- Some claims could be further clarified. For example, it would be suitable if the authors explained in more detail what kind of conduct is precisely encompassed in the term “nudging” during the provision of information to the patient. Another example might be the following claim: “scenarios which lacked patient involvement were generally associated with a significantly greater desire to complain than scenarios where the patient was well-informed and participated in decision making“ (p. 9). There is one scenario with very limited patient involvement, one SDM scenario, and three scenarios where the patient is informed by the physician but does not participate in SDM (see Figure 1). The difference between these “intermediate” scenarios seems to consist in the physician’s nudging rather than the level of patient involvement (which is the same). From this perspective, it is unclear what exactly do authors mean when writing about “scenarios which lacked patient involvement” and “scenarios where the patient was well-informed”.

Reviewer #5: Thanks to the authors for the opportunity to read this manuscript. I give my comments:

1. Abstract. It should be clearly indicated that these were mock clinical encounter using 30 scenarios.

2. Abstract. The authors write „…with various levels of patient involvement, decisions, and outcomes…”. Please add information about the decisions the authors have in mind.

3. Abstract. „With neutral involvement or nudging in favor of intervention, the desire to complain depended highly on the decision reached and on the patient’s course” – please explain what decision the authors mean.

4. Abstract. „…even in worst case scenarios” – please provide the result that supports this conclusion.

5. Introduction. The beginning of the introduction is very interesting, so it's a pity that the introduction is not longer. I am convinced that readers other than me would be equally eager to explore this issue. Please consider indicating at the end of the introduction the tested hypotheses.

6. Please consider including the first part of the discussion in the introduction.

7. Introduction. „The aim of our study was to explore the likely impact of patient involvement on health care users’ intentions to initiate a malpractice complaint given different approaches to patient involvement, while taking into account decisions made and healthcare outcomes.” – again, please explain what decision the authors mean.

8. Introduction. What other factors that influence a patient's complaints tendency have been investigated so far? How does the presented research complement this state of knowledge? In my opinion, the justification for this subject should be better justified.

9. Methods. No indication of the data collection period.

10. Please describe the mock clinical encounter in more detail

11. Methods. How many patients were assigned to specific scenarios? Did these groups differ in terms of sociodemographic variables?

12. Please consider including the sections „THEORETICAL FRAMEWORK: PATIENT INVOLVEMENT IN HEALTH CARE DECISIONMAKING” and „INTERVENTION – THE CLINICAL MODEL” in the introduction.

13. In my opinion the content of the "STATISTICAL ANALYSIS" section should mostly be found in the methods as a description of the study design / Study size.

14. In the "STATISTICAL ANALYSIS" section, there is no indication of what statistical program was used, what tests were performed, what significance level was used.

15. Results. „Irrespective of the level of patient involvement, respondents’ intent to complain increased with less favourable course scenarios (Likert ratings going up; please see Table 1)” - was it statistically significant?

16. Results. „Additionally, accross all types of decision-making, the inclination towards complaining was influenced by the decision made” - please provide the result of the statistical analysis supporting this claim.

17. As required by the journal, access to more detailed data should be provided, e.g. in addition to summary statistics, there should be data points behind medians.

Reviewer #6: This is a well-intended and thorough study evaluating impact of shared decision making on future patient dissatisfaction, up to and including malpractice complaints. As a result, authors use outcomes of Vignette Surveys as a hypothetical measurement of malpractice liability of Danish physicians who provide care ranging from neutral, “nudges” toward or against care, and SDM. However, as someone outside of the Danish (and European) community, the way the paper is written currently, I struggled to understand the rationale behind the Vignette Survey and claims authors tried to interpret from the results.

Comments from each section are listed below:

Abstract: list percentage of respondents instead of “6,756 men responded”

Introduction:

1. Authors begin discussing the ethical considerations of SDM in medical practice but don’t pursue this discussion in the context of Survey results or later discussion.

2. Authors must add more background regarding decision making and its impact on the patient physician relationship before jumping to claim that it is the physician’s responsibility to avoid malpractice liability or limit it to the best of their ability through inclusive strategies such as SDM. Other similar studies conducted in the literature are also not mentioned. Example: Braddock et al. discuss the role of the patient in clinical decisions – supporting foundational literature such as this is missing from the introduction (https://jamanetwork.com/journals/jama/fullarticle/192233).

3. Why were physicians not surveyed on their attitudes toward SDM? Authors chose to set-up the study design for male patients to indicate likelihood of reporting malpractice against their physician depending on several factors which this experimental vignette survey evaluates but the rationale is not clearly explained. This makes it seem as though care is delivered to prevent malpractice when a physician's duty is to provide optimal care - certain patients may prefer autonomy or rely on their doctor's medical expertise but this survey raises good concern that a patient's circle of care should involve SDM

4. It is interesting that the intro very briefly mentions that the "fear" of malpractice liability impacts quality of healthcare delivered. The relationship between SDM and this claim is not instantly apparent.

5. this intro makes me believe that the surveys would evaluate if patient participation in their treatment and care regimens would increase patient satisfaction and reduce malpractice complaints – specifically those that are FALSE accusations of malpractice... but if the malpractice is there - then it would be defined as malpractice against a patient? Later the authors mention that in Denmark, patients can initiate a malpractice suit and the survey questions are coded specific to that system. This should be mentioned in the intro, not the methods.

6. Paragraph 3, sentence 1: “health care users”, use the term patient for consistency

Methods:

1. There are 5 subheadings, PLOS ONE limits to 3 subheadings.

2. PSA needs to be defined in the first use of the term.

3. Figure 1 is not a robust framework as it is depicted here as a “continuum”. A continuum is a spectrum ranging from one opposite end to the other. Authors describe SDM as the “high involvement end” but nudging either in favor of the intervention or against is also more involvement than the neutral provision of information at the middle of the continuum. Type of “involvement” should be defined by the authors.

4. How did authors decide in the Vignettes when info shared by physician was tainted by personal preference or defensive strategies? And how can a patient recognize when expert medical advice for their condition is being shared versus these “defensive” strategies? An explanation will add depth and meaning to the paper

5. Intervention – clinical model: Paragraph 1, sentence 4 missing citation.

6. Survey Vignettes: Survey design needs to be explained further. Although hypothetical, certain scenarios are not described accurately enough to interpret the results of the survey as evidence of an impact of SDM on preventing malpractice complaints. For instance, if a clinician nudges (which the authors describe not as medical advice but personal preference or strategic decision) to get PSA screening, but patient chooses not to and later is diagnosed with untreatable PCa. The PSA screening was described earlier as having high rate of adverse events and relatively little gain. Again, the rationale behind this Study design is missing. Is this paper investigating validity of malpractice complaints or serve as a call to action for the medical community to involve SDM in patient care because that is optimal care? It needs to be clearer that authors believe SDM is optimal care and good care for patients prevents malpractice complaints as seen by the Survey results, whether complaints prevented are true or false.

Discussion: Interpretation of findings

1. paragraph 1, sentence 3, avoid describing findings as “this is a remarkable finding” and put it in context instead.

2. paragraph 1, sentence 4, “communication which fell below the accepted standard for informed consent”. There is very little discussion about informed consent, what standards are acceptable and how informed consent relates to SDM.

3. Paragraph 1, sentence 9, “SDM is superior to involving patients through provision of ‘neutral information’”. This is a big claim - neutral information I guess is medical advice? Tie in findings with the literature. In the intro authors described that not all patients are comfortable engaging in SDM but rather rely on doctors to tailor patient care to their medical profile – e.g. recommend screening if genetic histroy of PCa.

4. Paragraph 1, last sentence, “risks patients are interested in are not always the ones practitioners routinely disclose”. This is then a discussion about informed consent – the common ethical principle that all reasonable risks and benefits of a healthcare intervention or screening ought to be disclosed when engaging in a patient-physician discussion. Is an absence of informed consent resolved by SDM?

A limitation is this paper is not likely generalizable to others since hypothetical scenarios are specific to male Danish patients and their physicians. The impact of the paper may be stronger if authors cite the actual number of malpractice complaints filed against physicians in Denmark to highlight the importance of their Survey. Further open and medically informative discussion is needed for optimal patient care and satisfaction.

Overall, this is an insightful paper that picks up on ethical aspects of a common patient-physician encounter and if comments provided can be addressed, this would be an important paper for SDM in countries where complaints are filed frequently, burdening healthcare providers.

6. PLOS authors have the option to publish the peer review history of their article (what does this mean?). If published, this will include your full peer review and any attached files.

Reviewer #1: No

Reviewer #2: No

Reviewer #3: No

Reviewer #4: No

Reviewer #5: No

Reviewer #6: No

---

## [Author Response · Author response to Decision Letter 0]

10 May 2021

Cover letter giving a point-by-point response to the concerns, manuscript number: PONE-D-20-40761: “Does shared decision-making in healthcare reduce malpractice complaints?” submitted to PLOS ONE

We thank PLOS ONE for insightful and constructive criticism on our manuscript. We appreciate this opportunity to revise our manuscript and optimize its content. 

The manuscript has now been revised. This cover letter gives an overview on revisions and a point-by-point response to the issues raised in editors’ and reviewers’ reports. 

Response to editors’ comments

By way of introduction, editor states that “1. Please ensure that your manuscript meets PLOS ONE's style requirements, including those for file naming. The PLOS ONE style templates can be found at https://journals.plos.org/plosone/s/file?id=wjVg/PLOSOne_formatting_sample_main_body.pdf andhttps://journals.plos.org/plosone/s/file?id=ba62/PLOSOne_formatting_sample_title_authors_affiliations.pdf. ”

Authors’ comment: 

Again, we wish to thank editors for giving us an opportunity to revise the manuscript. In the revision, we have changed our manuscript’s style to ensure that it meets PLOS ONE's style requirements, including those for file naming.

Editors afterwards add:

“2. In your Methods section, please provide additional information about the participant recruitment method and the demographic details of your participants. Please ensure you have provided sufficient details to replicate the analyses such as: a) the recruitment date range (month and year), b) a table of relevant demographic details. ”

Authors’ comment: 

Thanks for this comment. We now provide precise information about the recruitment method in a separate subsection of the ‘Materials and methods’ section called ‘Recruitment of participants’. Basic demographic characteristics of our sample have been reported before (please see table in Birkeland et al. BMC Health Services Research (2020) 20:851), and we have now added this information in the methods section (under ‘Recruitment of participants’). 

Editors afterwards add:

“3. Please include additional information regarding the survey or questionnaire used in the study and ensure that you have provided sufficient details that others could replicate the analyses. For instance, if you developed a questionnaire as part of this study and it is not under a copyright more restrictive than CC-BY, please include a copy, in both the original language and English, as Supporting Information. ”

Authors’ comment: 

This is a helpful suggestion, thank you. We now include a copy of the questionnaire, in both the original language (Danish) and English, as Supporting Information.

Editors afterwards add:

“4. We note that you have indicated that data from this study are available upon request. PLOS only allows data to be available upon request if there are legal or ethical restrictions on sharing data publicly. For information on unacceptable data access restrictions, please see http://journals.plos.org/plosone/s/data-availability#loc-unacceptable-data-access-restrictions. ”

Authors’ comment: 

We have now uploaded an anonymized minimal data set, including all data necessary to replicate our study findings, to a stable, public repository and provide PLOS One with the relevant URLs, DOIs, and accession numbers. 

Editors afterwards add:

“In your revised cover letter, please address the following prompts: a) If there are ethical or legal restrictions on sharing a de-identified data set, please explain them in detail (e.g., data contain potentially identifying or sensitive patient information) and who has imposed them (e.g., an ethics committee). Please also provide contact information for a data access committee, ethics committee, or other institutional body to which data requests may be sent. ”

Authors’ comment: 

We have now uploaded an anonymized minimal data set, including all data necessary to replicate our study findings, to a stable, public repository and provide PLOS One with the relevant URLs, DOIs, and accession numbers. Please see just above.

Editors afterwards add:

“b) If there are no restrictions, please upload the minimal anonymized data set necessary to replicate your study findings as either Supporting Information files or to a stable, public repository and provide us with the relevant URLs, DOIs, or accession numbers. Please see http://www.bmj.com/content/340/bmj.c181.long for guidelines on how to de-identify and prepare clinical data for publication. For a list of acceptable repositories, please see http://journals.plos.org/plosone/s/data-availability#loc-recommended-repositories. We will update your Data Availability statement on your behalf to reflect the information you provide. ”

Authors’ comment: 

Thanks for this comment. We have now uploaded an anonymized minimal data set, necessary to replicate our study findings, to a stable, public repository and provide PLOS One with the relevant URLs, DOIs, and accession numbers. Please see just above.

Editors afterwards add:

“5. Your ethics statement should only appear in the Methods section of your manuscript. If your ethics statement is written in any section besides the Methods, please move it to the Methods section and delete it from any other section. Please ensure that your ethics statement is included in your manuscript, as the ethics statement entered into the online submission form will not be published alongside your manuscript. ”

Authors’ comment: 

Thanks for this comment. The ethics statement now appears only in the ‘Methods’ section.

Response to Reviewer 1’s comments

By way of introduction, reviewer 1 states that “Thanks to the authors for the opportunity to read this manuscript. I give my comments: 1. Abstract. It should be clearly indicated that these were mock clinical encounter using 30 scenarios. ”

Authors’ comment: 

 We agree. In the revised abstract’s ‘Methods’ section it is now clearly indicated that vignettes use scenarios to illustrate mock clinical encounters.

Reviewer afterwards adds:

“2. Abstract. The authors write „…with various levels of patient involvement, decisions, and outcomes…”. Please add information about the decisions the authors have in mind. ”

Authors’ comment: 

Thanks for the comment. This is now clearly indicated in the abstract. Please see under the ‘Methods’ section.

Reviewer afterwards adds:

“3. Abstract. „With neutral involvement or nudging in favor of intervention, the desire to complain depended highly on the decision reached and on the patient’s course” – please explain what decision the authors mean. ”

Authors’ comment: 

Thanks for this important comment. The abstract has been revised and now provides information about the decision (to have a PSA test or not).

Reviewer afterwards adds:

“4. Abstract. „…even in worst case scenarios” – please provide the result that supports this conclusion. ”

Authors’ comment: 

Thanks for the comment. This has been now rephrased.

Reviewer afterwards adds:

“5. Introduction. The beginning of the introduction is very interesting, so it's a pity that the introduction is not longer. I am convinced that readers other than me would be equally eager to explore this issue. Please consider indicating at the end of the introduction the tested hypotheses.”

Authors’ comment: 

Thanks for the helpful suggestion. The introduction has been now further elaborated. Also, at the end of the introduction, we now indicate the tested hypothesis. 

Reviewer afterwards adds:

“6. Please consider including the first part of the discussion in the introduction. ”

Authors’ comment: 

As suggested, we have now moved some of the discussion to the introduction. Please see revised manuscript.

Reviewer afterwards adds:

“7. Introduction. „The aim of our study was to explore the likely impact of patient involvement on health care users’ intentions to initiate a malpractice complaint given different approaches to patient involvement, while taking into account decisions made and healthcare outcomes.” – again, please explain what decision the authors mean. ”

Authors’ comment: 

Thanks for this important comment. We have now specified that the decision is about having an intervention (PSA test) or not. Please see revised manuscript.

Reviewer afterwards adds:

“8. Introduction. What other factors that influence a patient's complaints tendency have been investigated so far? How does the presented research complement this state of knowledge? In my opinion, the justification for this subject should be better justified. ”

Authors’ comment: 

Thanks for this comment. We have now added information about other factors that may influence a patient's tendency to complain. Please see revised manuscript, the ‘Introduction’.

Reviewer afterwards adds:

“9. Methods. No indication of the data collection period. ”

Authors’ comment: 

Thanks for the comment. In the revision, information about the data collection period now is provided. Please see under ‘Recruitment of participants’.

Reviewer afterwards adds:

“10. Please describe the mock clinical encounter in more detail.”

Authors’ comment: 

In the revision, we have described the mock clinical encounter in more detail. Please see also the new ‘Table 1’.

Reviewer afterwards adds:

“11. Methods. How many patients were assigned to specific scenarios? Did these groups differ in terms of sociodemographic variables? ”

Authors’ comment: 

We have now included the numbers of patients assigned to specific scenarios. According to previously published analyses, these groups did not differ in terms of sociodemographic variables. 

Reviewer afterwards adds:

“12. Please consider including the sections „THEORETICAL FRAMEWORK: PATIENT INVOLVEMENT IN HEALTH CARE DECISIONMAKING” and „INTERVENTION – THE CLINICAL MODEL” in the introduction. ”

Authors’ comment: 

Thanks for the comment. As there are so few tested theories linking patient engagement in health care decision making to malpractice liability, we see this study as more ”inductive” research to develop theory rather than deductive research to test existing theory. Hence, we think that figure 1 and ‘Clinical model..’ descriptions both represent templates set up by the research team to further investigate health users’ view on participation and are not as such theories which are agreed on according to the background research literature. 

Reviewer afterwards adds:

“13. In my opinion the content of the "STATISTICAL ANALYSIS" section should mostly be found in the methods as a description of the study design / study size. ”

Authors’ comment: 

Thanks for the comment. We have slightly changed the sub-heading in the methods section to ‘Study size and statistical analyses’. Please see revised manuscript. 

Reviewer afterwards adds:

“14. In the "STATISTICAL ANALYSIS" section, there is no indication of what statistical program was used, what tests were performed, what significance level was used. ”

Authors’ comment: 

We now have added the statement “All statistical analyses were performed using Stata 16” to this section.

Reviewer afterwards adds:

“15. Results. „Irrespective of the level of patient involvement, respondents’ intent to complain increased with less favourable course scenarios (Likert ratings going up; please see Table 1)” - was it statistically significant? ”

Authors’ comment: 

Good point. We now have formally tested these differences for statistical significance, and added the P-value to the manuscript (statistically significant). 

Reviewer afterwards adds:

“16. Results. „Additionally, accross all types of decision-making, the inclination towards complaining was influenced by the decision made” - please provide the result of the statistical analysis supporting this claim. ”

Authors’ comment: 

As suggested, we have now formally tested this assertion, and added the P-value to the manuscript. 

Reviewer afterwards adds:

“17. As required by the journal, access to more detailed data should be provided, e.g. in addition to summary statistics, there should be data points behind medians. ”

Authors’ comment: 

Thanks for the comment. More detailed data are now provided. Please see revised manuscript.

Response to Reviewer 2’s comments

By way of introduction, reviewer 2 states that “I thank the authors for the very interesting read. The manuscript identifies a gap in the literature on shared decision-making (SDM) in clinical settings in that it points out that there is insufficient data on the correlation between SDM and the prevalence of malpractice complaints. The authors addressed this gap by conducting a web-based survey with a random national sample of men living in Denmark. Survey participants were presented with one of 30 vignettes describing hypothetical scenarios of experiences with prostate specific antigen screening (PSA). The vignettes differed in the level of patient-involvement, whether a PSA was undertaken, and clinical outcome (whether the participant developed prostate cancer and whether they were successfully treated). The survey elicited participants’ stated desire to seek compensation or complain about the physician’s care for the scenario with which they were presented. The manuscript’s main contribution lies in analyzing the relationship between different hypothetical levels of patient involvement in PSA screening and participants’ stated desire to complain.

Overall, the manuscript’s structure is well constructed and the text is clearly written. The current state of the literature is reviewed concisely, yet sufficiently. The chosen methodology is sound and appropriate for purpose and the presented approach builds on previous research. The results presented in the manuscript are relevant and add to the discipline’s body of knowledge. The study’s limitations are addressed clearly. Some minor issues remain which can easily be addressed by the authors. ”

Authors’ comment: 

We thank Reviewer 2 for this very positive feedback on our manuscript. 

Reviewer afterwards adds:

“1) Abstract: “Conclusions: Involving patients in decision-making appears to offer some prevention against malpractice complaints with patients engaged in SDM less likely to complain even in worst case scenarios”. The phrasing is unfortunate as it presents malpractice complaints simply as something to be prevented, regardless of their merit. The phrase could be misread as saying that the likelihood of a patient complaining bears no relation to the quality of care received. The argument presented in the main text is much more nuanced and avoids these misinterpretations. I suggest rephrasing.”

Authors’ comment: 

Thanks for this important comment. The research team has rephrased the conclusion of the ‘Abstract’ and the conclusion of the manuscript to align with the more nuanced perspective presented in the body of the paper. 

Reviewer afterwards adds:

“2) P.3: “Some clinicians and researchers have questioned this assumption. They contend that choice of interventions is fundamentally a health care provider’s responsibility and that, rather than participating in decision-making, patients look for provider authenticity and willingness to assume responsibility for their medical care (2, 3). From this perspective, the onus is on clinicians to avoid malpractice liability, fueling the practice of ‘defensive medicine’ for fear of being criticized for failing to order tests or other interventions prior to a poor outcome (4)”. It is not immediately clear how the second sentence follows from the first. The incentive structure for clinicians might be shaped as described in the latter sentence, but the authors need to spell out how this results from the mentioned shift in perspective. ”

Authors’ comment: 

Thanks! Reviewer 2 is entirely correct about this observation. These sentences have now been rephrased. 

Reviewer afterwards adds:

“3) p.3: “While patients’ wishes to take legal action may inherently vary with the outcomes of care, empirical studies recurrently report deficient communication about options, risks, and benefits to be an important predictor of malpractice complaints (5-7). Correspondingly, it would be natural to hope that greater information sharing and patient participation in decision-making may prevent some complaints, but evidence is lacking (7)” This is an important argument that connects the quality of received care to the likelihood to complain. Some version of it should be added to the abstract to compliment the clinician-centered perspective presented there. ”

Authors’ comment: 

Thanks for this important comment. The argument is now added to the abstract. 

Reviewer afterwards adds:

“4) p.3: “STUDY OVERSIGHT” should read “study overview”. ”

Authors’ comment: 

Agreed. This has been revised.

Reviewer afterwards adds:

“5) p.3.: “We conducted a large national survey on a randomly selected sample of adult men in Denmark”. Decision to elicit preferences of the general population instead of the affected should briefly be motivated. ”

Authors’ comment: 

Thanks for this perceptive comment. The decision is now briefly explained. Please see revised ‘Study overview’ and amendment in the ‘Limitations’ section.

Reviewer afterwards adds:

“6) p.3: “The survey was developed with patient and public involvement from the gender and age group of interest”. Please briefly expand on the methodology used for patient involvement. ”

Authors’ comment: 

Thanks for the comment. We have briefly expanded on the methodology used for patient involvement in survey development. 

Reviewer afterwards adds:

“7) P.5: “We used a web-based survey (RedCap®) and invited a large random national sample of men living in Denmark through the Danish National authorities’ web-portal for communication with citizens (9)”. Add a brief discussion of potential selection effects here. ”

Authors’ comment: 

Thanks for pointing out this matter. We have elaborated further on this concern in the limitations section with description of the selection effects potentially arising when using a web-based survey approach. 

Reviewer afterwards adds:

“8) P.8: Figure 3: Light yellow on white extremely hard to read, consider changing color scheme. ”

Authors’ comment: 

The color scheme in Figure 3 has been changed

Reviewer afterwards adds:

“9) P.9: The first paragraph of the discussion section should be integrated into the background section. ”

Authors’ comment: 

In the revised manuscript, we have tried to integrate parts of the first paragraph into the background section. Please also see above.

Reviewer afterwards adds:

“10) P.9-11 Just a musing but it might be worthwhile to conceptualize patient involvement as an important aspect of good quality care rather than as something that happens independent from the health intervention proper. This may or may not be worth mentioning in the conclusion. In any case, it would provide a good counter-balance to the physician-centered view currently starring prominently in the manuscript. ”

Authors’ comment: 

We agree. In the revised manuscript, as suggested by the reviewer, we have conceptualized patient involvement as an important aspect of good quality care (please also see conclusion).

Response to Reviewer 3’s comments

By way of introduction, reviewer 3 states that “The submitted manuscript describes a large national survey on a randomly selected sample of adult men in Denmark. The studied aimed to analyze whether higher levels of patient involvement in decision-making about healthcare interventions are likely to prevent malpractice complaints. The manuscript is well written and the methodology of the study is well described. The findings are relevant and well described in relation to previous studies. No fundamental objections to this manuscript, but I would recommend to expand the introduction, which is too short at this moment. ”

Authors’ comment: 

Thanks for the comment. In the revised manuscript, the introduction has been expanded. Please also refer to comments made by reviewer 2 and changes made in response to those comments. 

Reviewer afterwards adds:

“It is unclear from the manuscript whether the data is publicly available. Considering that it is an anonymous dataset, it shouldn’t be a problem to deposit the dataset. ”

Authors’ comment: 

We have now uploaded a anonymized minimal data set including all data necessary to replicate our study findings to a stable, public repository and provide PLOS One with the relevant URLs, DOIs, and accession numbers. Please also see our response to editor’s comment above.

Response to Reviewer 4’s comments

By way of introduction, reviewer 4 states that “The paper deals with a very relevant and up-to-date question of the effects of shared decision-making (SDM) on the patients’ inclination to complain or sue healthcare providers. While it is often presumed that there is a negative correlation between the two phenomena, it is yet to be proven. The paper provides a valuable contribution to the recent discussion which is not only very interesting for other researchers but also potentially practical for healthcare providers, health professionals, and various policymakers. For the reasons listed here and below, the reviewer recommends acceptation of the paper (perhaps with very minor revisions).” 

Authors’ comment: 

Reviewer 4 must be thanked for this very kind comment.

Reviewer afterwards adds:

“The methods used are appropriate for the aim of the study. Due to its statistically limited benefits, relatively high risk of adverse effects, and the possible grave negative consequences of false-negative results, the Prostate Specific Antigen screening seems to be a very suitable model of medical intervention for the purposes of the study. As far as the reviewer is qualified to assess statistical analysis’s adequacy, the collection, analysis, and interpretation of the data were accurate. The figures and tables are clear, readable, and support the results of the study.

The respondent sample is big enough to draw a conclusion. Technical aspects regarding the representativeness of the sample, as well as the patient and public participation in the development of the survey and some other data, are described and evaluated in already-published papers in different journals, which are referred to in the reviewed paper.

Inclusion of three variables (i.e. not only patient participation in decision-making but also the decision made by the patient and the outcome) and analysis of their interrelations arguably made the results more reliable since in the real world, the said variables (among others) undoubtedly affect the patient’s decision to complain. Nevertheless, the study is still limited by its exclusive use of hypothetical judgments, which are arguably not influenced by emotions related to one’s own (perceived) health harm. Even though the respondents reported feeling able to identify with the patients in the case scenarios, further research would be necessary to establish the importance of emotions and other aspects of decision-making regarding complaining or suing healthcare providers in the real world. Nevertheless, this and several other limitations are addressed in the paper in a way that can be considered sufficient. ”

Authors’ comment: 

Reviewer 4 must be thanked for this very insightful comment.

Reviewer afterwards adds:

“There are just two minor issues: - Rarely there are typing errors (e.g. “litterature on p. 5 or “coloumn” on p. 6). It is only a marginal problem, but another proofreading might be useful. ”

Authors’ comment: 

Thanks for the comment. In the revised manuscript, we have corrected typing errors, and done an extra proofreading. 

Reviewer afterwards adds:

“- Some claims could be further clarified. For example, it would be suitable if the authors explained in more detail what kind of conduct is precisely encompassed in the term “nudging” during the provision of information to the patient. Another example might be the following claim: “scenarios which lacked patient involvement were generally associated with a significantly greater desire to complain than scenarios where the patient was well-informed and participated in decision making“ (p. 9). There is one scenario with very limited patient involvement, one SDM scenario, and three scenarios where the patient is informed by the physician but does not participate in SDM (see Figure 1). The difference between these “intermediate” scenarios seems to consist in the physician’s nudging rather than the level of patient involvement (which is the same). From this perspective, it is unclear what exactly do authors mean when writing about “scenarios which lacked patient involvement” and “scenarios where the patient was well-informed”. ”

Authors’ comment: 

Thanks for these important comments. In the revision, we have explained a little further what we mean by ‘nudging’. Please see revised manuscript under ‘A study model on …’. Furthermore, ‘well-informed’ has been changed to ‘received plain, neutral information’. Please see the ‘Discussion’ section under ‘Interpretation of findings’. In total, six scenarios out of the 30 possibly scenarios illustrated courses of care which lacked patient involvement (patient received no information and physician made the decision about doing a PSA test; please see figure 1). When comparing those six scenarios to their neutral information ‘counterparts’, the desire to complain generally was greater. 

Response to Reviewer 5’s comments

This reviewer’s comments seem to be a copy of reviewer 1’s comments

Response to Reviewer 6’s comments

By way of introduction, reviewer 6 states that “This is a well-intended and thorough study evaluating impact of shared decision making on future patient dissatisfaction, up to and including malpractice complaints. As a result, authors use outcomes of Vignette Surveys as a hypothetical measurement of malpractice liability of Danish physicians who provide care ranging from neutral, “nudges” toward or against care, and SDM. However, as someone outside of the Danish (and European) community, the way the paper is written currently, I struggled to understand the rationale behind the Vignette Survey and claims authors tried to interpret from the results. ”

Authors’ comment: 

Thanks for the comment. We are sorry that the reviewer struggled to understand the rationale behind the vignette survey and claims we tried to interpret from our results. We have now made a thorough revision in order to clarify matters. Among other edits, we have changed the title of the paper in order to prevent misunderstanding of its aim. Please see the revised manuscript. 

Reviewer afterwards adds:

“Comments from each section are listed below: Abstract: list percentage of respondents instead of “6,756 men responded” ”

Authors’ comment: 

Thanks for the comment. The percentage of respondents is now listed in the revised abstract. 

Reviewer afterwards adds:

“Introduction: 1. Authors begin discussing the ethical considerations of SDM in medical practice but don’t pursue this discussion in the context of Survey results or later discussion.”

Authors’ comment: 

Thanks for this important comment. The ethical dimension is now pursued in the context of survey results in the discussion. Please see revised manuscript.

Reviewer afterwards adds:

“2. Authors must add more background regarding decision making and its impact on the patient physician relationship before jumping to claim that it is the physician’s responsibility to avoid malpractice liability or limit it to the best of their ability through inclusive strategies such as SDM. Other similar studies conducted in the literature are also not mentioned. Example: Braddock et al. discuss the role of the patient in clinical decisions – supporting foundational literature such as this is missing from the introduction (https://jamanetwork.com/journals/jama/fullarticle/192233). ”

Authors’ comment: 

Thanks for this important comment. In the revised ‘Introduction’ section, more background information is provided including reference to the study referred to by reviewer 6.

Reviewer afterwards adds:

“3. Why were physicians not surveyed on their attitudes toward SDM? Authors chose to set-up the study design for male patients to indicate likelihood of reporting malpractice against their physician depending on several factors which this experimental vignette survey evaluates but the rationale is not clearly explained. This makes it seem as though care is delivered to prevent malpractice when a physician's duty is to provide optimal care - certain patients may prefer autonomy or rely on their doctor's medical expertise but this survey raises good concern that a patient's circle of care should involve SDM. ”

Authors’ comment: 

Thanks for this important comment. Rather than strictly focusing on SDM, the scope of the paper is to study whether higher levels of patient involvement in decision-making about healthcare interventions are likely to reduce malpractice complaints. Therefore, the title may be misleading and raise the expectation that the paper would more specifically investigate attitudes toward SDM. Therefore, the title has been now changed to more closely reflect the paper’s aim. Furthermore, we have re-written parts of the ‘Introduction’, ‘Discussion’, and ‘Conclusion’ sections to more clearly indicate that the aim is not to prevent malpractice but to provide optimal care.

Reviewer afterwards adds:

“4. It is interesting that the intro very briefly mentions that the "fear" of malpractice liability impacts quality of healthcare delivered. The relationship between SDM and this claim is not instantly apparent. ”

Authors’ comment: 

Thanks for the comment. We have tried to elaborate on the relationship between greater patient involvement and quality of healthcare in the revised manuscript. Please see ‘Introduction’, ‘Discussion’, and ‘Conclusion’ sections.

Reviewer afterwards adds:

“5. this intro makes me believe that the surveys would evaluate if patient participation in their treatment and care regimens would increase patient satisfaction and reduce malpractice complaints – specifically those that are FALSE accusations of malpractice... but if the malpractice is there - then it would be defined as malpractice against a patient? Later the authors mention that in Denmark, patients can initiate a malpractice suit and the survey questions are coded specific to that system. This should be mentioned in the intro, not the methods. ”

Authors’ comment: 

Legally, “malpractice’ requires both deviation from the standard of care and harm resulting from that deviation. Practically, malpractice is “in the eye of the beholder.” The authors believe the spectrum of physician PSA behaviors described in these scenarios are all within the standard of care based on physicians’ varying practices. However, individual people considering themselves in these scenarios may disagree. Our approach was to document participants’ reactions to the various combinations of physician PSA screening behaviors and patient outcomes. We did not use scenarios where the physician behavior and outcomes would more clearly reflect malpractice, such as an elevated PSA not being noticed leading to a prostate cancer death. We have, however, provided a brief explanation of the different ways of complaining about health care (‘collectively referred to as “complaints”’) to the introduction. 

Reviewer afterwards adds:

“6. Paragraph 3, sentence 1: “health care users”, use the term patient for consistency. ”

Authors’ comment: 

Thanks for the comment. We have removed ‘health care users’ in sentence 1 in the ‘Study size and statistical analyses’ subsection. Throughout, we have tried to be consistent in our use of ‘patients’ rather than ‘health care users’.

Reviewer afterwards adds:

“Methods: 1. There are 5 subheadings, PLOS ONE limits to 3 subheadings. ”

Authors’ comment: 

According to submission guidelines, no more than three levels can be used but there seems to be no limit on the number of sub-headings (so-called level 2 headings). Please also see, e.g., Jeppesen et al. Short-term spheroid culture of primary colorectal cancer cells as an in vitro model for personalizing cancer medicine. PLoS One. 2017 Sep 6;12(9):e0183074. doi: 10.1371/journal.pone.0183074 (10 level 2 headings).

Reviewer afterwards adds:

“2. PSA needs to be defined in the first use of the term. ”

Authors’ comment: 

Thanks for this comment. We have now made sure that PSA is defined with the first use of the term.

Reviewer afterwards adds:

“3. Figure 1 is not a robust framework as it is depicted here as a “continuum”. A continuum is a spectrum ranging from one opposite end to the other. Authors describe SDM as the “high involvement end” but nudging either in favor of the intervention or against is also more involvement than the neutral provision of information at the middle of the continuum. Type of “involvement” should be defined by the authors. ”

Authors’ comment: 

Thanks for this comment. In the revision, ‘continuum’ and ‘spectrum’ descriptions have been changed to description of different ‘levels’. Please see revised manuscript and revised figure 1.

Reviewer afterwards adds:

“4. How did authors decide in the Vignettes when info shared by physician was tainted by personal preference or defensive strategies? And how can a patient recognize when expert medical advice for their condition is being shared versus these “defensive” strategies? An explanation will add depth and meaning to the paper. ”

Authors’ comment: 

Thanks for the comment. As it is described in the ‘Materials and methods’ section of the paper, the decision to have a PSA is preference sensitive. Hence, because of the high risk of adverse effects relative to the limited gain, it tends to be considered an individual decision whether the possible benefits are deemed to outbalance the risks associated with the test and ensuing treatment. All participants subject to vignettes illustrating neutral or nudged information received the same ‘core’ information about the PSA test. The only communication distinguishing ‘neutral’ from ‘nudged’ involvement was whether the doctor would – or would not – recommend a PSA screening test. We have tried to clarify this in more detail in the revised manuscript’s ‘Materials and methods’ section, subsection ‘Survey vignettes’.

Reviewer afterwards adds:

“5. Intervention – clinical model: Paragraph 1, sentence 4 missing citation. ”

Authors’ comment: 

Thanks for the comment. The citation has been now inserted.

Reviewer afterwards adds:

“6. Survey Vignettes: Survey design needs to be explained further. Although hypothetical, certain scenarios are not described accurately enough to interpret the results of the survey as evidence of an impact of SDM on preventing malpractice complaints. For instance, if a clinician nudges (which the authors describe not as medical advice but personal preference or strategic decision) to get PSA screening, but patient chooses not to and later is diagnosed with untreatable PCa. The PSA screening was described earlier as having high rate of adverse events and relatively little gain. Again, the rationale behind this Study design is missing. Is this paper investigating validity of malpractice complaints or serve as a call to action for the medical community to involve SDM in patient care because that is optimal care? It needs to be clearer that authors believe SDM is optimal care and good care for patients prevents malpractice complaints as seen by the Survey results, whether complaints prevented are true or false. ”

Authors’ comment: 

Thanks for the comment. The title of the paper has been changed in order to not be misleading regarding the study’s purpose. Additionally, much more information about the survey vignettes are now provided in the manuscript and reference is made to a full version of the questionnaire. Please also see response to editor’s comment and response to reviewer 6’s comment number 3. 

Reviewer afterwards adds:

“Discussion: Interpretation of findings. 1. paragraph 1, sentence 3, avoid describing findings as “this is a remarkable finding” and put it in context instead. ”

Authors’ comment: 

Thanks for the comment. In the revision, the “this is a remarkable finding” statement has been removed.

Reviewer afterwards adds:

“2. paragraph 1, sentence 4, “communication which fell below the accepted standard for informed consent”. There is very little discussion about informed consent, what standards are acceptable and how informed consent relates to SDM. ”

Authors’ comment: 

Thanks for the comment. This has been rephrased and the relationship between IC and SDM is further elaborated in the ‘Theroretical framework’ subsection of the ‘Materials and methods’ section of the revised manuscript.

Reviewer afterwards adds:

“3. Paragraph 1, sentence 9, “SDM is superior to involving patients through provision of ‘neutral information’”. This is a big claim - neutral information I guess is medical advice? Tie in findings with the literature. In the intro authors described that not all patients are comfortable engaging in SDM but rather rely on doctors to tailor patient care to their medical profile – e.g. recommend screening if genetic histroy of PCa. ”

Authors’ comment: 

Thanks for the comment. ‘Neutral information’ simply is provision of basic information about a procedure that can be chosen in the patient’s situation without any recommendation or further guidance about whether the procedure would be the right choice for that particular patient. In the revised manuscript, more information is provided about what is mean by ‘neutral information’, etc. Please see revision’s ‘Materials and methods’ section, the ‘A study model on ...’ and ‘Case vignettes’ subsections. Actually, analyses suggest that – from a complaint perspective- SDM is superior to involving patients through provision of ‘neutral information’. This has been further clarified in the revised manuscript’s ‘Discussion’ section, subsection ‘Interpretation of findings’.

Reviewer afterwards adds:

“4. Paragraph 1, last sentence, “risks patients are interested in are not always the ones practitioners routinely disclose”. This is then a discussion about informed consent – the common ethical principle that all reasonable risks and benefits of a healthcare intervention or screening ought to be disclosed when engaging in a patient-physician discussion. Is an absence of informed consent resolved by SDM? ”

Authors’ comment: 

The authors see SDM as an evolution and extension of informed consent, involving not only providing accurate information about an intervention’s harms, benefits, and alternatives, but also meaningful patient involvement in making a decision to proceed.

Reviewer afterwards adds:

“A limitation is this paper is not likely generalizable to others since hypothetical scenarios are specific to male Danish patients and their physicians. The impact of the paper may be stronger if authors cite the actual number of malpractice complaints filed against physicians in Denmark to highlight the importance of their Survey. Further open and medically informative discussion is needed for optimal patient care and satisfaction. ”

Authors’ comment: 

Thanks for the comment. Reviewer 6 is right that the possibility exists that the study is not generalizable to other countries, but we are not sure that such a lack of generalizability is likely. It might be common in many countries that men see their doctor, sometimes are faced with the decision to have a PSA test and some men are diagnosed with prostate cancer. Danish men’s reactions to the scenarios we present may generalize to men in other Western countries given the international controversy about the appropriate role of PSA screening, even if the malpractice climate differs from country to country. The scenarios presented to participants certainly involve physician behaviors that might be seen in the United States, for example. We have acknowledged the limitation and possible lack of generalizability of a study done in a single country.

Reviewer afterwards adds:

“Overall, this is an insightful paper that picks up on ethical aspects of a common patient-physician encounter and if comments provided can be addressed, this would be an important paper for SDM in countries where complaints are filed frequently, burdening healthcare providers. ”

Authors’ comment: 

Thanks for the comment. In the revision, we have aimed to address the reviewer’s comments.

Again, we wish to express our thanks for constructive criticism and comments.

Please do not hesitate to contact in case of any questions or concerns.

Yours sincerely, 

Søren Birkeland /May 6, 2021

---

## [Decision Letter · Decision Letter 1]

9 Jun 2021

PONE-D-20-40761R1

Does greater patient involvement in healthcare decision-making affect malpractice complaints? –A large case vignette survey

PLOS ONE

Dear Dr. Birkeland,

Thank you for submitting your manuscript to PLOS ONE. After careful consideration, we feel that it has merit but does not fully meet PLOS ONE’s publication criteria as it currently stands. Therefore, we invite you to submit a revised version of the manuscript that addresses the points raised during the review process.

We look forward to receiving your revised manuscript.

Kind regards,

Prof. Ritesh G. Menezes, M.B.B.S., M.D., Diplomate N.B.

Academic Editor

PLOS ONE

Journal Requirements:

Reviewers' comments:

Reviewer's Responses to Questions

**Comments to the Author**

1. If the authors have adequately addressed your comments raised in a previous round of review and you feel that this manuscript is now acceptable for publication, you may indicate that here to bypass the “Comments to the Author” section, enter your conflict of interest statement in the “Confidential to Editor” section, and submit your "Accept" recommendation.

Reviewer #1: All comments have been addressed

Reviewer #2: All comments have been addressed

Reviewer #3: All comments have been addressed

Reviewer #6: All comments have been addressed

2. Is the manuscript technically sound, and do the data support the conclusions?

Reviewer #1: Yes

Reviewer #2: Yes

Reviewer #3: Yes

Reviewer #6: Yes

3. Has the statistical analysis been performed appropriately and rigorously? 

Reviewer #1: Yes

Reviewer #2: I Don't Know

Reviewer #3: Yes

Reviewer #6: Yes

4. Have the authors made all data underlying the findings in their manuscript fully available?

Reviewer #1: Yes

Reviewer #2: Yes

Reviewer #3: Yes

Reviewer #6: Yes

5. Is the manuscript presented in an intelligible fashion and written in standard English?

Reviewer #1: Yes

Reviewer #2: Yes

Reviewer #3: Yes

Reviewer #6: Yes

6. Review Comments to the Author

Reviewer #1: (No Response)

Reviewer #2: I would like to thank the authors for the very interesting manuscript. They have addressed all of my comments to my full satisfaction.

Reviewer #3: No further comments. All comments have been addressed by an extensive answer to the various questions.

Reviewer #6: I thank the authors for their patience waiting for this peer review as well as their continued determination to present their manuscript in its best possible shape. All previous comments/concerns have been addressed sufficiently, suggested papers have been referenced, and I now recommend that this paper proceed, given a few minor revisions, for publication in PLOS ONE.

The authors made an effort to not only describe the findings of their study, as was done in the original manuscript submission, but also have now worked toward setting their results within the larger context of the field of patient-centered care and decision making. Limitations are well acknowledged and the need for this paper to inform clinical decision-making strategies in the context of complaint reduction and delivery of improved care that is ethically sound is established.

This study will make a valuable contribution to the literature on patient-physician encounters, patient-centered care and patient autonomy in clinical decision making. The authors responded to my reviewer comment stating they “see SDM as an evolution and extension of informed consent, involving not only providing accurate information about an intervention’s harms, benefits, and alternatives, but also meaningful patient involvement in making a decision to proceed.” I find that this statement would be well suited in the conclusion of the paper with respect to the findings.

Another few very minor comments and suggestions:

INTRODUCTION:

1. Paragraph 1, last sentence – “well-established” is a strong word, especially since only one citation is provided per claim. Suggestion: “it is acknowledged that rate of…”. It would also be worthwhile to understand these claims. For instance, you can explain that these claims support that higher risk medical procedures tend to correspond with higher rates of complaints.

2. Paragraph 3, rephrase suggestion: The aim of our study was to explore "how different approaches to patient involvement impact patients' intentions to initiate a complaint, while accounting for the decision made and the subsequent health outcomes of that decision."

3. Paragraph 3, the null hypothesis tests the “urge to complain”. "Urge" is quite ambiguous - a concrete outcome like "initiation or filing of a healthcare complaint" may be better.

MATERIALS and METHODS

1. Study overview: paragraph 1, sentence 2 “from as wide a group of men as possible” is a good addition. This is also to ensure outcomes are generalizable to a larger percent of the population.

2. Study overview: paragraph 1 sentence 3 – “(please see below)” could be more specific, e.g., (Fig 1)

3. A study model on patient involvement in health care decision-making: paragraph 1, sentence 1: add period instead of comma before “Pertinent”.

4. A study model on patient involvement in health care decision-making: paragraph 1, sentence 3: “attentive” should be replaced by “attention”

5. Intervention – the clinical model: paragraph 1, sentence 6 add [based on] – “…PSA screening tends to be considered an individual…decision [based on] whether…”.

6. Survey Vignettes: paragraph 2, last sentence states an average of two ratings was used. A brief explanation of why you did not separate ratings of those inclined to seek compensation and those who simply seek to complain would be helpful.

RESULTS

1. Paragraph 1, sentence 5 – “Likert ratings going up” can be reworded “as observed by an increase in Likert ratings reported, which reflect increasing likelihood of complaining”

DISCUSSION

1. Paragraph 1, sentence 1 suggested rephrase: "Patient involvement and considerations about an individual's right to autonomy is a common discussion in bioethics. Moreover, patient-centered care has become an essential part of delivering quality care in modern medicine"

2. Paragraph 1, last sentence: replace “urge” with “likelihood”

3. Interpretation of Findings: paragraph 1, sentence 10: “…risks that [patients may prioritize more than clinicians] and those risks…”

4. Interpretation of Findings: paragraph 2, sentence 6: “our results are interesting” is overtly descriptive of your own results. Suggested rephrase: "Viewing from a medico-legal perspective, in our survey, scenarios with PSA screening...etc."

5. Overall, when interpreting your findings, please add in parentheses the p-values and related CIs whenever you mention trends toward association of significant or not significant results from Table 2 and Table 3

LIMITATIONS

1. Paragraph 1, sentence 4: “Anyway” replaced by “however” and add citation regarding the controversy surrounding PSA screening

2. Paragraph 1, last sentence: cite the other previous research

CONCLUSIONS

1. Second last sentence: “urge” replaced by “likelihood” to complain, as observed by Likert scores

2. Last sentence: “Overall, [greater patient involvement, such as SDM,] achieved…regardless of the eventual [health] outcome”.

Congratulations, Dr. Birkeland et al. on this important and welcome contribution to the medico-legal literature on patient involvement, clinical decision-making and complaints!

7. PLOS authors have the option to publish the peer review history of their article (what does this mean?). If published, this will include your full peer review and any attached files.

Reviewer #1: No

Reviewer #2: No

Reviewer #3: No

Reviewer #6: No

---

## [Author Response · Author response to Decision Letter 1]

15 Jun 2021

Cover letter giving a point-by-point response to the concerns, manuscript number: PONE-D-20-40761R1: “Does greater patient involvement in healthcare decision-making affect malpractice complaints? –A large case vignette survey” submitted to PLOS ONE

Once again, we thank PLOS ONE for insightful and constructive criticism on our manuscript. We appreciate this opportunity to re-revise our manuscript and optimize its content. 

We are happy that reviewers 1, 2, and 3 are all satisfied with our revision. This cover letter gives an overview on revisions and a point-by-point response to the issues raised in reviewer 6’s report. 

Response to Reviewer 6’s comments

By way of introduction, reviewer 6 states that “I thank the authors for their patience waiting for this peer review as well as their continued determination to present their manuscript in its best possible shape. All previous comments/concerns have been addressed sufficiently, suggested papers have been referenced, and I now recommend that this paper proceed, given a few minor revisions, for publication in PLOS ONE.”

Authors’ comment: 

We thank Reviewer 6 for this very positive feedback on our manuscript. 

Reviewer afterwards adds:

“The authors made an effort to not only describe the findings of their study, as was done in the original manuscript submission, but also have now worked toward setting their results within the larger context of the field of patient-centered care and decision making. Limitations are well acknowledged and the need for this paper to inform clinical decision-making strategies in the context of complaint reduction and delivery of improved care that is ethically sound is established.

This study will make a valuable contribution to the literature on patient-physician encounters, patient-centered care and patient autonomy in clinical decision making. The authors responded to my reviewer comment stating they “see SDM as an evolution and extension of informed consent, involving not only providing accurate information about an intervention’s harms, benefits, and alternatives, but also meaningful patient involvement in making a decision to proceed.” I find that this statement would be well suited in the conclusion of the paper with respect to the findings. ”

Authors’ comment: 

We thank reviewer 6 for this very positive comment and for the suggestion. We have now added the statement to the conclusion of the paper.

Reviewer afterwards adds:

“Another few very minor comments and suggestions: INTRODUCTION: 1. Paragraph 1, last sentence – “well-established” is a strong word, especially since only one citation is provided per claim. Suggestion: “it is acknowledged that rate of…”. It would also be worthwhile to understand these claims. For instance, you can explain that these claims support that higher risk medical procedures tend to correspond with higher rates of complaints.”

Authors’ comment: 

Thanks for this comment and suggestion. We have now changed “well-established” to “acknowledged” and provide the explanation suggested by reviewer 6. 

Reviewer afterwards adds:

“2. Paragraph 3, rephrase suggestion: The aim of our study was to explore "how different approaches to patient involvement impact patients' intentions to initiate a complaint, while accounting for the decision made and the subsequent health outcomes of that decision."”

Authors’ comment: 

Thanks for the comment. This has been now rephrased.

Reviewer afterwards adds:

“3. Paragraph 3, the null hypothesis tests the “urge to complain”. "Urge" is quite ambiguous - a concrete outcome like "initiation or filing of a healthcare complaint" may be better.”

Authors’ comment: 

Thanks for the comment. This has been now rephrased. 

Reviewer afterwards adds:

“MATERIALS and METHODS 1. Study overview: paragraph 1, sentence 2 “from as wide a group of men as possible” is a good addition. This is also to ensure outcomes are generalizable to a larger percent of the population.”

Authors’ comment: 

Thanks for the comment. We agree. We have now added this rationale. 

Reviewer afterwards adds:

“2. Study overview: paragraph 1 sentence 3 – “(please see below)” could be more specific, e.g., (Fig 1) ”

Authors’ comment: 

Thanks for the comment. We have now specified this.

Reviewer afterwards adds:

“3. A study model on patient involvement in health care decision-making: paragraph 1, sentence 1: add period instead of comma before “Pertinent””

Authors’ comment: 

Thanks for this comment. We have added period instead of comma before “Pertinent”.

Reviewer afterwards adds:

“4. A study model on patient involvement in health care decision-making: paragraph 1, sentence 3: “attentive” should be replaced by “attention””

Authors’ comment: 

Thanks for the suggestion. We have replaced “attentive” by “attention”.

Reviewer afterwards adds:

“5. Intervention – the clinical model: paragraph 1, sentence 6 add [based on] – “…PSA screening tends to be considered an individual…decision [based on] whether…”.”

Authors’ comment: 

Thanks for the suggestion. We have added “based on” in ‘Intervention – the clinical model’: paragraph 1, sentence 6.

Reviewer afterwards adds:

“6. Survey Vignettes: paragraph 2, last sentence states an average of two ratings was used. A brief explanation of why you did not separate ratings of those inclined to seek compensation and those who simply seek to complain would be helpful.”

Authors’ comment: 

Thanks for comment. We have now specified that we used the average to provide an overall estimate of the complaint likelihood and we decided not to investigate the two ratings separately, as the distinction between compensation claims and complaints vary among countries.

Reviewer afterwards adds:

“RESULTS. 1. Paragraph 1, sentence 5 – “Likert ratings going up” can be reworded “as observed by an increase in Likert ratings reported, which reflect increasing likelihood of complaining””

Authors’ comment: 

Thanks for the comment. This has been rephrased in agreement with reviewer’s suggestion. 

Reviewer afterwards adds:

“DISCUSSION. 1. Paragraph 1, sentence 1 suggested rephrase: "Patient involvement and considerations about an individual's right to autonomy is a common discussion in bioethics. Moreover, patient-centered care has become an essential part of delivering quality care in modern medicine"”

Authors’ comment: 

Thanks for this comment. We have rephrased the first paragraph in accordance with reviewer’s suggestion. Please see revised manuscript, beginning of ‘Discussion’. 

Reviewer afterwards adds:

“2. Paragraph 1, last sentence: replace “urge” with “likelihood””

Authors’ comment: 

Thanks for the comment. This has been rephrased. 

Reviewer afterwards adds:

“3. Interpretation of Findings: paragraph 1, sentence 10: “…risks that [patients may prioritize more than clinicians] and those risks…””

Authors’ comment: 

Thanks for the comment. We have corrected this. Please see revised manuscript, “Interpretation of Findings”: paragraph 1, sentence 10. 

Reviewer afterwards adds:

“4. Interpretation of Findings: paragraph 2, sentence 6: “our results are interesting” is overtly descriptive of your own results. Suggested rephrase: "Viewing from a medico-legal perspective, in our survey, scenarios with PSA screening...etc."”

Authors’ comment: 

Thanks for the comment. We agree! This has been now corrected. Please see revised manuscript, “Interpretation of Findings”: paragraph 2, sentence 6. 

Reviewer afterwards adds:

“5. Overall, when interpreting your findings, please add in parentheses the p-values and related CIs whenever you mention trends toward association of significant or not significant results from Table 2 and Table 3.”

Authors’ comment: 

Thanks for the comment. We have now added in parentheses the p-values and related CIs when mentioning trends toward association of significant or not significant results from Table 2 and Table 3. 

Reviewer afterwards adds:

“LIMITATIONS. 1. Paragraph 1, sentence 4: “Anyway” replaced by “however” and add citation regarding the controversy surrounding PSA screening”

Authors’ comment: 

Thanks for this comment. In Paragraph 1, sentence 4, we replaced “Anyway” by “however” and added a citation regarding the controversy surrounding PSA screening. 

Reviewer afterwards adds:

“2. Paragraph 1, last sentence: cite the other previous research.”

Authors’ comment: 

Thanks for this comment. In Paragraph 1, last sentence, we now cite the other previous research.

Reviewer afterwards adds:

“CONCLUSIONS. 1. Second last sentence: “urge” replaced by “likelihood” to complain, as observed by Likert scores.”

Authors’ comment: 

Thanks for this comment. The second last sentence has been rephrased.

Reviewer afterwards adds:

“2. Last sentence: “Overall, [greater patient involvement, such as SDM,] achieved…regardless of the eventual [health] outcome”.”

Authors’ comment: 

Thanks for this comment. The last sentence has been rephrased.

Reviewer finally adds:

“Congratulations, Dr. Birkeland et al. on this important and welcome contribution to the medico-legal literature on patient involvement, clinical decision-making and complaints!”

Authors’ comment: 

We thank Reviewer 6 for this very positive feedback on our manuscript. 

More generally, we would like to express our thanks again for very insightful and constructive criticism, comments and suggestions.

Please note that we added the foot note information to tables 2 and 3 that N amounts to 6,755 as one participant did not respond to complaint likelihood items.

Please do not hesitate to contact in case of any questions or concerns.

Yours sincerely, 

Søren Birkeland /June 15, 2021

---

## [Editor Report · Decision Letter 2]

21 Jun 2021

Does greater patient involvement in healthcare decision-making affect malpractice complaints? –A large case vignette survey

PONE-D-20-40761R2

Dear Dr. Birkeland,

We’re pleased to inform you that your manuscript has been judged scientifically suitable for publication and will be formally accepted for publication once it meets all outstanding technical requirements.

Kind regards,

Prof. Ritesh G. Menezes, M.B.B.S., M.D., Diplomate N.B.

Academic Editor

PLOS ONE

---

## [Editor Report · Acceptance letter]

25 Jun 2021

PONE-D-20-40761R2 

Does greater patient involvement in healthcare decision-making affect malpractice complaints? –A large case vignette survey 

Dear Dr. Birkeland:

I'm pleased to inform you that your manuscript has been deemed suitable for publication in PLOS ONE. Congratulations! Your manuscript is now with our production department. 

Kind regards, 

on behalf of

Prof. Dr. Ritesh G. Menezes 

Academic Editor

PLOS ONE